# From viability to cell death: Claims with insufficient evidence in high-impact cell culture studies

**Ali Burak Özkaya**[1]\*, **Caner Geyik**[2,3]

1 Department of Medical Biochemistry, Faculty of Medicine, İzmir University of Economics, İzmir, Turkey,
2 Department of Medical Biochemistry, Faculty of Medicine, İstinye University, İstanbul, Turkey, 3 ISUMKAM Molecular Cancer Research Center, İstinye University, Istanbul, Turkey

\* ali.ozkaya@ieu.edu.tr

## Abstract

### Background

Reliability of preclinical research is of critical concern. Prior studies have demonstrated the low reproducibility of research results and recommend implementing higher standards to improve overall quality and robustness of research. One understudied aspect of this quality issue is the harmony between the research hypotheses and the experimental design in published work.

### Methods and findings

In this study we focused on highly cited cell culture studies and investigated whether commonly asserted cell culture claims such as viability, cytotoxicity, proliferation rate, cell death and apoptosis are backed with sufficient experimental evidence or not. We created an open access database containing 280 claims asserted by 103 different high-impact articles as well as the results of this study. Our findings revealed that only 64% of all claims were sufficiently supported by evidence and there were concerning misinterpretations such as considering the results of tetrazolium salt reduction assays as indicators of cell death or apoptosis.

### Conclusions

Our analysis revealed a discordance between experimental findings and the way they were presented and discussed in the manuscripts. To improve quality of pre-clinical research, we require clear nomenclature by which different cell culture claims are distinctively categorized; materials and methods sections to be written more meticulously; and cell culture methods to be selected and utilized more carefully. In this paper we recommend a nomenclature for selected cell culture claims as well as a methodology for collecting evidence to support those claims.

**Citation:** Özkaya AB, Geyik C (2022) From viability to cell death: Claims with insufficient evidence in high-impact cell culture studies. PLoS ONE 17(2): e0250754. https://doi.org/10.1371/journal.pone.0250754

**Data Availability Statement:** All relevant data are within the manuscript and its Supporting Information files.

**Funding:** The author(s) received no specific funding for this work.

**Competing interests:** The authors have declared that no competing interests exist.

## Introduction

An alarming concern exists regarding reliability of the published research findings [1, 2]. This is particularly evident in preclinical studies, as clinical translation is minimal [3]. Such low efficiency in research has been discussed extensively in recent years and the lack of reproducibility and overall quality are agreed upon as the main culprits of the problem [4]. Reproducibility in preclinical research is estimated to be between 10% to 25% [5, 6] and the cost of irreproducible research is calculated to be at least 28 billion USD/year in USA alone [7]. There are many factors contributing to this crisis, including lack of robustness, biased design, use of inadequate models (cell line and/or animal), underpowered studies (insufficient sample size), lack of proper controls (i.e. samples that are expected to produce [positive control] or expected not to produce results [negative control], poor use of statistics and the absence of replication/confirmation studies [8]. It is important to note that these design problems often extend to questionable research practices such as p-hacking (performing multiple different statistical analyses just to obtain significance) and cherry-picking (concealing inconsistent or contradictory findings).

Scientists agree that the standards for publishing preclinical research must be raised in such a way as to encourage robustness and rigor [6, 9]. Therefore, many aspects of the preclinical study design have been tackled by various studies over the years [7, 8]. However, the question of whether we can trust results of published preclinical studies remain at large. Even though understanding key concepts and methods for reporting data has been suggested as critical to preserving scientific findings [10], one important aspect of the process, the compatibility of the way the manuscript was written with the actual experimental design, has been overlooked. More specifically, the relationship between the claims of the studies and the evidence provided to support these claims remains underexamined. In this study we investigated if the evidence provided by high-impact studies sufficiently supports cell culture claims authors asserted in their manuscript. It is important to note that we focused only on selected commonly used cell culture claims and our analysis does not cover all the asserted claims. In many cases cell culture may be a small part of the study with minimal effect to its conclusions. Accordingly, our findings should not be interpreted as a measure of article quality.

While we were trying to select the claims to include in our analysis, one of the first things we have noticed was the inconsistency in the nomenclature. Many claims such as cytotoxicity, viability, growth, and proliferation were used interchangeably by the authors. Moreover, there were several publications in which only one type of evidence (tetrazolium reduction assay results) was provided to support various claims. When we searched the literature to refine the consensus nomenclature, much to our disappointment, we could not find any. Possibly, many of these terms were not considered uncommon, unfamiliar, or vague enough to be defined in high-impact reviews or guidelines, or to be included in the glossary sections of the molecular biology, biochemistry and even cell culture textbooks. Therefore, in the current work, we propose a series of definitions and recommendations mostly based on different sections of "Guidance Document on Good In Vitro Method Practices" by OECD [11] which was the only document we find that might be considered as a consensus nomenclature source. Based on these definitions, we analyzed high-impact cell culture studies to investigate if the asserted claims are justifiably backed with evidence or not.

## Methods

The study consisted of three phases. In phase one, we selected high-impact cell culture studies. In phase two, we identified some of the cell culture claims asserted by the authors as well as the evidence provided by them to support these claims. In the final phase, we analyzed the sufficiency of the evidence for each of the claims.

## Article selection

We searched Web of Science (WOS) database (Clarivate Analytics) for studies that contain at least one of these keywords: "cytotoxicity, viability, cell death, growth inhibition, proliferation, or anti-cancer". We included original research articles using *in vitro* techniques. The search string below was used in advanced search feature of WOS:

WOS core database (TS = ("cytotoxicity" OR "viability" OR "cell death" OR "growth inhibition" OR "growth inhibitory" OR "proliferation" OR "anti cancer") AND TS = ("cell culture" OR "in vitro" OR "cell line")) AND LANGUAGE:(English) AND DOCUMENT TYPES: (Article)

Studies published in 2017 and 2018 were retrieved in 10.03.2020. We exported the data as an Excel file, sorted the list based on citations received and selected the most cited 121 publications (each receiving at least 65 citations) as high-impact studies for further analysis. Upon investigation, we excluded 18 articles, including those not containing a claim, those not carried out in cell culture, and those that were review articles and perspectives. After the exclusion process, 103 studies remained for claim analysis.

## Claim selection and definitions

Using any one of six distinct terms to present or discuss a finding in the studies was considered a claim. We identified these as: *proliferation rate*, *viability*, *cell death*, *apoptosis*, *cytotoxicity*, and *cell growth*. We outlined our definitions for these claims as follows by using OECD guide GIVIMP [11]:

*Proliferation rate* represents how fast a group of cells divide over time. To provide sufficient evidence for proliferation rate, an end-point measurement must be able to differentiate the change in division capabilities of the cells from cell death. If the treatment of question induces cell death in the treatment group, there would be fewer living cells (compared to untreated control) without a decrease in proliferation rate. Accordingly, methods specifically focusing on replication rate (such as nucleotide incorporation) or measuring the number of viable cells without a treatment over time as well as real-time observations, and proliferation markers were considered as sufficient evidence.

*Viability* represents the number of living cells. It is the broadest term since there is no specification regarding the factor affecting the number (such as proliferation rate or cell death). Any method directly measuring the number of living cells and methods measuring metabolic activity or total protein content were considered as sufficient evidence for viability.

Since *cell death*, by definition, requires cells to die, end point analysis measuring the abundance of living cells cannot provide sufficient evidence for this claim as the measurement does not differentiate the decrease in number due to dying cells from slowed-down proliferation rate. In some cases, when the viability of treated cells is so low that the decrease in proliferation rate is not sufficient to explain the viability loss, cell-death can be assumed. However, even in those cases it is impossible to detect individual contributions of cell-death and decreased proliferation rate to reduced viability. Therefore, only assays measuring death-related alterations such as membrane integrity loss or cell-death specific markers were considered to provide sufficient evidence.

*Apoptosis* is a form of programmed cell death which has well established and characterized distinct features. The methods that can demonstrate changes in features such as exposure of phosphatidylserine on the outer membrane, DNA fragmentation, morphological changes

(nuclear condensation and membrane blebbing) and molecular switches (caspase activation via cleavage) resulting in apoptosis were considered as sufficient evidence to show apoptosis.

*Cytotoxicity* indicates being toxic to cells. Being toxic itself is a broad term and there are conflicting definitions in use. We decided to consider it as cell death instead of decreased viability as the most widely accepted capability of a toxic agent is killing (as in cytotoxic T cells and cytotoxic chemotherapy), and accepted evidence indicating cell death as sufficient to prove cytotoxicity. This decision had an impact on the final analysis as many of the articles might have used the term to represent viability decrease. We addressed this in results section.

*Cell growth* may indicate either proliferation rate or cell size change depending on the definition embraced. Since, there already is a term representing proliferation rate as the name implies, we first considered to accept growth as a measure of increased cell volume. However, after investigating the articles in our list, we realized that the term was exclusively used to indicate proliferation rate and consequently we embraced that definition in our analysis.

## Database construction

We constructed a database in Airtable to carry out evidence analysis. Information from WOS database including "article name", "DOI", "citation count", "journal name" as well as our parameters of interest including "claim", "evidence", "method", "sufficiency of evidence", and "subject area" were entered for every article investigated.

Here, "method" represents scientific methods used in the study whereas "evidence" is defined as a supergroup of methods measuring same biological phenomenon. For example, two separate methods such as lactate dehydrogenase (LDH) activity assay and PI both of which measure membrane damage as an indicator of cell death were classified into "membrane integrity" evidence supergroup. Similarly, various tetrazolium and resazurin reduction assays were considered to provide "dehydrogenase activity" evidence as an indicator of cellular metabolic activity.

We have also divided the studies in two notional groups of "subject area" based on field information provided by WOS. The first one is "Biochemistry, Molecular Biology, Genetics, and Medicine" and the second one is "Chemistry, Chemical/Biomedical Engineering, and Materials Science".

The database is accessible via the link: https://airtable.com/shrClTE87e1l28ExG and its contents are also accessible as S1 Data (Database CSV spread sheet).

## Evidence analysis

Evidence was analyzed for each of the claims using a case-by-case approach. We refer to the definitions we have embraced and the OECD guide GIVIMP [11] to determine if the measured parameter provides sufficient evidence for that claim. Table 1 summarizes the claims asserted and the evidence provided by the articles we investigated. The table also provides information regarding our decision on whether a type of evidence is sufficient for a claim or not as well as the rationale behind that decision.

If there were multiple types of evidence for a single claim, we focused on the strongest of that evidence. For example, even though expression of Bcl-2 members (such as Bax/Bcl-2 ratio) is commonly investigated along with other markers of apoptosis, it is classified as insufficient because it does not provide proper evidence for apoptosis by itself. However, if an established feature of apoptosis (like DNA fragmentation) was demonstrated along with Bcl-2 expression results, we focused only on that evidence, classified it as sufficient and ignored the rest. Another example to this is citing viability assay results as evidence for cell death induction or proliferation rate change when there is further evidence. Normally by itself, viability assay

**Table 1. Claims and the evidence provided by the articles along with the sufficiency of the evidence and the rationale for our decision.**

| Claim | Evidence | Methods/Assays Used | Sufficient | Rationale |
|---|---|---|---|---|
| **Apoptosis** | Caspase-3 activity | Caspase-3 activity assay, Cleaved caspase-3 protein measurement | Yes | Molecular switch resulting in apoptosis |
| | Cell morphology | Electron microscopy | Yes | Established feature/characteristic of apoptosis |
| | DNA fragmentation | Acridine orange staining, TUNEL assay | Yes | Established feature/characteristic of apoptosis |
| | Phosphatidylserine exposure | Annexin-V staining | Yes | Established feature/characteristic of apoptosis |
| | Dehydrogenase activity | Tetrazolium reduction assay | No | Indicator of viability but not apoptosis |
| | Membrane integrity | Propidium iodide staining | No | Indicator of cell death but not apoptosis |
| | Bcl-2 Family expression | Protein expression measurement | No | Indicator of a pro-apoptotic signal but not of the apoptosis itself. |
| **Cell Death** | Membrane integrity | Staining by: DAPI, Ethidium bromide, Ethidium homodimer-1, LDH activity, Propidium iodide, Zombie UV | Yes | Indicates dying/death cells |
| | Phosphatidylserine exposure | Annexin-V staining | Yes | Established feature of apoptosis and accordingly cell death |
| | Cell count | Real-time cell imaging | No | No distinction between cell-death and slowed-down proliferation rate |
| | Dehydrogenase activity | Tetrazolium reduction assay | No | |
| | Esterase activity | Calcein AM staining | No | |
| **Cytotoxicity** | Membrane integrity | Staining by: 7-AAD, Ethidium homodimer-1, Hemolysis assay, LDH activity, Propidium iodide | Yes | Indicates dying/death cells |
| | Phosphatidylserine exposure | Annexin-V staining | Yes | Established feature of apoptosis and accordingly cell death |
| | Dehydrogenase activity | Tetrazolium reduction assay, Resazurin reduction assay | No | No distinction between cell-death and slowed-down proliferation rate |
| **Cell Growth** | Cell count | Real-time cell imaging | Yes | Indicates cell division and proliferation |
| | Colony formation | Crystal violet assay | Yes | Indicates cell division and proliferation |
| | Nucleotide incorporation | BrdU assay | Yes | Indicates cell division and proliferation |
| | Dehydrogenase activity | Tetrazolium reduction assay | No | No distinction between cell-death and slowed-down proliferation rate |
| Claim | Evidence | Methods/Assays Used | Sufficient | Rationale |
| **Proliferation** | Dye inclusion | Celltrace violet assay | Yes | Dye dilution over generations indicates proliferation |
| | Ki67 expression | Expression at protein level | Yes | Direct marker for cell proliferation |
| | Nucleotide incorporation | $[H^3]$-Thymidine, BrdU, EdU | Yes | Direct marker for cell proliferation via replication |
| | Phospho-histone H3 | Anti-Ph3 staining | Yes | Lack of Ph3 is an indicator of non-proliferating cells |
| | Cell count | Cell analyzer, Cell and particle counter, Unknown | Conditional | Since these methods cannot identify the reason for viability change, it is considered sufficient only if there is no treatment that may cause cell death. In all other cases it is considered insufficient. |
| | Colony formation | Crystal violet assay | Conditional | |
| | Dehydrogenase activity | Tetrazolium reduction assay, Resazurin reduction assay | Conditional | |
| | DNA amount | Picogreen staining | Conditional | |
| | Actin | Phalloidin staining | No | Actin visualization is not an indicator of proliferation |
| | Esterase activity | Fluorescein DA staining | No | No distinction between cell-death and slowed-down proliferation rate |
| | Signaling Pathways | Protein expression measurement | No | May indicate activation of growth-related signaling pathways but does not provide evidence for proliferation rate. |

(*Continued*)

**Table 1.** (Continued)

| Viability | | | | |
|---|---|---|---|---|
| | ATP amount | Luciferase-based assay | Yes | ATP amount correlates with the number of viable cells |
| | Dehydrogenase activity | Tetrazolium reduction assay, Resazurin reduction assay | Yes | DH activity correlates with the number of viable cells |
| | Dye inclusion | Staining by: Acridine orange, Hoechst, GhostDye | Yes | Indicates cells with intact cell membrane |
| | Esterase activity | Calcein AM staining, Fluorescein DA staining | Yes | Indicates cells with intact cell membrane and active esterase |
| | Total protein | Sulforhodamine B assay | Yes | Indicates viable cells |
| | Cell count | Cell Analyzer, MAP2 staining, Unknown | Conditional | Imaging of living cells were considered sufficient. If the method was not mentioned in the article, it was considered as insufficient evidence. |
| | Membrane integrity | Staining by: 7-AAD, Propidium iodide, Trypan blue | Conditional | Utilization of cell impermeable cytotoxicity dyes considered insufficient. (7-AAD and PI). |
| | | | | Trypan blue was an exception since it is possible to observe viable cells directly. |
| | Signaling Pathways | Protein expression measurement | No | May indicate activation of growth-related signaling pathways but does not provide evidence for viability. |
| | Unknown | N/A | No | N/A |

results do not provide information regarding to the cause of the viability change. However, if the mechanism was elucidated via further experimentation and those experiments were cited while making the claim, we then focused only on those experiments instead of the viability assay.

There were some cases where the same type of evidence was determined to be sufficient or insufficient based on the experimental design. Viability indicators such as cell count, colony formation, DNA amount, and dehydrogenase activity were considered to provide sufficient evidence for proliferation only if there was no treatment (hence no reason for a change in cell death ratio). Moreover, if the method utilized to obtain the evidence was not mentioned in the article (or in the S1 Data), we then classified it as insufficient (independent from the actual evidence) as was the case for cell count providing insufficient evidence for viability and proliferation [12].

Another such deviation was membrane integrity as evidence for viability. Normally we considered these cell-death markers (7-AAD and PI) as insufficient, due to the fact they do not indicate living cells by themselves. However, since it is not only possible but quite common to observe cells under microscope after trypan staining, we decided to consider it as sufficient evidence for viability.

## The effects of the journal and subject area on evidence sufficiency

After the sufficiency of evidence was assessed for each claim, we then sought to find whether being published in a specific journal or in a subject area would affect sufficiency rate. Seven journals were selected for analysis as they meet our criteria of having at least 10 claims, namely *Biomaterials*, *Cell*, *Cell Death Dis.*, *Mol. Cancer*, *Nature*, *Oncotarget*, *Ann. Biomed. Eng.* (96 claims out of 280, 34.28%). Every journal was compared to the complete data set excluding themselves. Two subject areas: "Biochemistry, Molecular Biology, Genetics, and Medicine" and "Chemistry, Chemical/Biomedical Engineering, and Materials Science" were compared to each other. Fisher's exact test (2-tail) were used in all comparisons.

## Results

We have investigated 280 claims asserted by 103 different high-impact articles. We identified 6 unique claims supported by 20 types of evidence all of which was obtained via 38 different

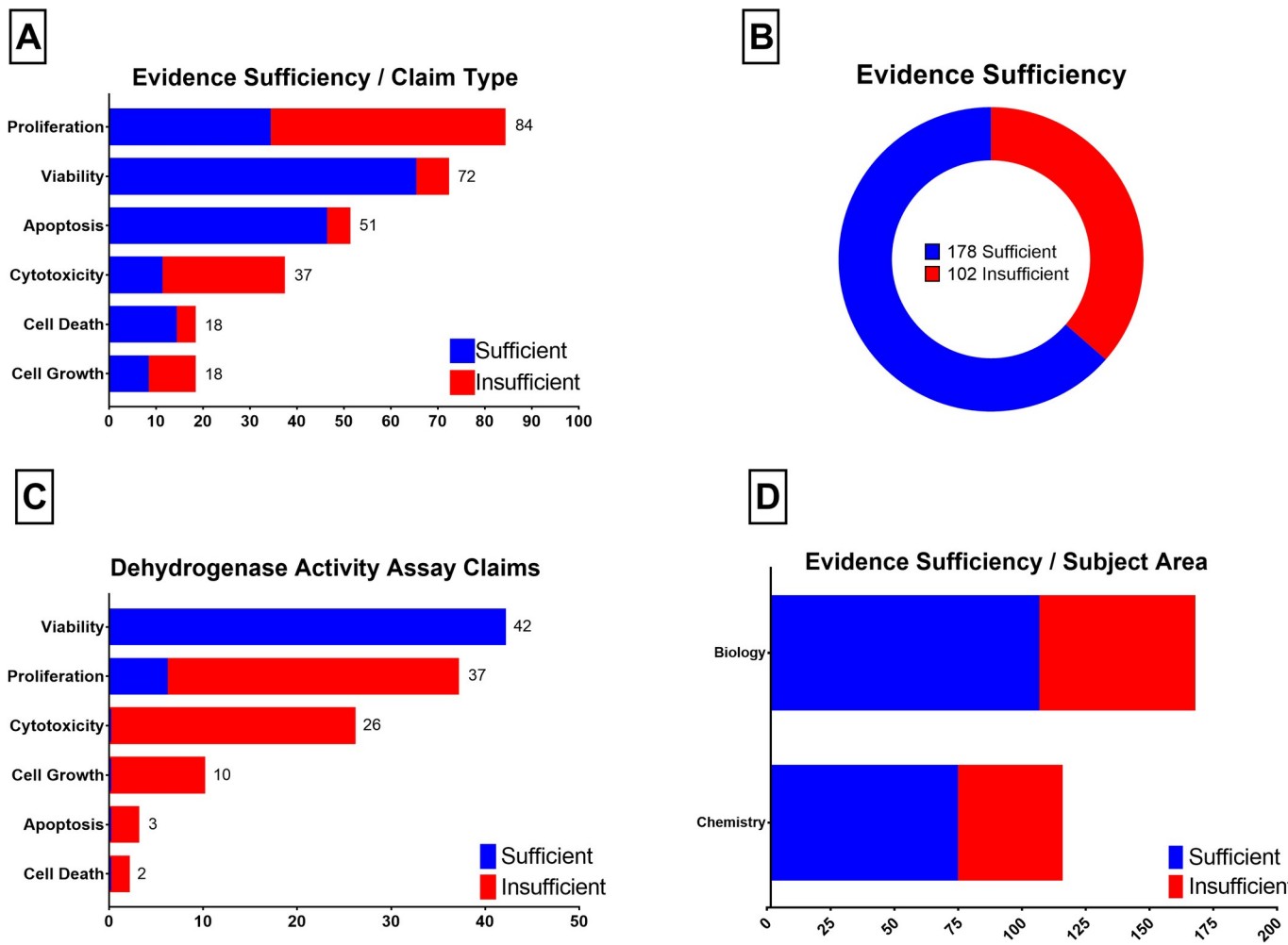

**Fig 1. Evidence sufficiency analysis results.** 1A. Evidence sufficiency according to the claim type; 1B. The number of claims with sufficient or insufficient evidence; 1C. Evidence sufficiency of claims supported by dehydrogenase activity assay; 1D. Evidence sufficiency according to the article subject area.

methods. Details are presented in the database (accessible via: https://airtable.com/shrClTE87e1l28ExG or as a S1 Data).

The most common claim was proliferation rate changes with 84 claims, followed by viability, apoptosis, cytotoxicity, cell death, and growth changes (Fig 1A). Upon investigation, we considered the evidence of 102 claims (36%), which was asserted by 67 different studies (65%) as insufficient (Fig 1B). Claims of cytotoxicity (11 sufficient in 37 total claims, 30%), proliferation (34 in 84, 40%), and cell growth (10 in 18 claims, 56%) particularly lacked proper evidence. Viability (64 in 71, 90%), apoptosis (46 in 51, 90%), and cell death (15 in 19, 79%), on the other hand, were more frequently claimed with proper evidence (Fig 1A).

In case of cytotoxicity, there is a possibility that the authors may have used the term to indicate decrease in cell viability instead of the definition we embraced (cell death). If we had considered cytotoxicity as a measure of viability instead of cell death, all the cytotoxicity claims would have had sufficient evidence increasing the overall sufficiency rate from 64% to 73%.

Measurement of dehydrogenase activity was by far the most common type of evidence. 120 different claims (43% of all claims) including viability, proliferation, cytotoxicity, cell growth, cell death, and even apoptosis put forward dehydrogenase activity findings as evidence

(Fig 1C). The most common method used to measure dehydrogenase activity was tetrazolium reduction assay supporting 114 different claims (41% of all claims). Measurement of dehydrogenase activity was followed by measurement of phosphatidylserine exposure (31 claims) and membrane integrity (28 claims).

While the results of dehydrogenase activity assays were interpreted correctly only in 48 claims (48 in 120, 40%), the second most common evidence, measurement of phosphatidylserine exposure, was interpreted correctly (as an indicator of apoptosis) in all related claims (31 in 31, 100%). Similarly, measurement of membrane integrity was mostly correctly utilized as evidence (25 in 28, 89%) (Fig 1C).

We also analyzed whether the subject area or the journal that the article was published in might be an indicator of evidence claim relationship. According to our analysis, neither the subject area (Fig 1D) nor the journal, have significant influence over evidence sufficiency.

## Discussion

There is a valid concern regarding reliability of preclinical research and many agree that we need strategies in place to improve the standards [6, 9]. There is a need for a comprehensive guideline for *in vitro* studies, which clearly and distinctively defines commonly used cell culture terms such as viability, cytotoxicity, proliferation, and cell death along with the correct methodology to measure them. Here, we offer a nomenclature and methodology recommendation for cell culture studies with such claims.

According to the definitions we embraced, viability represents the number of living cells and when there is a decrease in cell viability, there are two possible causes: decrease in how fast a group of cells divide over time (proliferation rate decrease) or induction of cell-death. Cytotoxicity represents reduction of viability via cell-death. Cytostaticity or inhibition of cell growth represents reduction of viability via decreased proliferation rate. Accordingly, methods measuring the number of living cells are appropriate for determining viability and methods detecting dying/dead cells are required to assert claims of cytotoxicity or cell-death. On the other hand, to claim a proliferation change, one must be able to determine the change in number of viable cells while taking cell-death into account.

In this study we created a database containing the data from highly-cited cell culture studies and analyzed the data based on our definitions. Since we focused only on the list of claims we have selected and defined, our analysis is neither an indicator of article quality nor an evaluation of all the claims of the articles. Our findings revealed a discordance between the cell culture claims and the evidence of these studies. This was especially evident in studies utilizing the findings of tetrazolium reduction assay, an indicator of cellular metabolic activity, alone to support various claims. Striking examples include article id#9 [13] claiming viability, proliferation, cytotoxicity and apoptosis changes, and article id#26 [14] claiming viability, proliferation, and growth changes with results from this assay. This is partly due to these assay kits being advertised by their manufacturers as a tool to measure viability, cytotoxicity, proliferation and growth. Combining this with being relatively easier to perform and affordable leads to these assays being perceived as a one-size-fits-all solution by research groups wishing to avoid more complicated cell culture methods. However, this reductionist approach makes it difficult for the findings obtained from the study (assay results) to provide a meaningful answer to the research question (does this treatment kill cells?) of the article. When this approach is embraced by a high impact work, its negative effects may extend beyond its own reliability.

In fact, the articles we analyzed were cited more than 9000 times as of December 2021 (within two to four years). Admittedly, claims without sufficient evidence may not be the reason for citation in most of the cases as cell culture may have a small part in the study. However,

the impact of such influential studies with unreliable findings on preclinical research is undeniably large. Defining distinct cell culture claims and adequate methodology to measure them is the first step to increase reliability of similar studies. However, that first step may not be enough if we ignore other issues surrounding basic cell culture techniques.

Even though, in this study we considered measurement of metabolic activity as an indicator of viability as it is by far the most preferred viability assay [15], measuring metabolic activity as an indicator of cell viability must be challenged. Formazan production in tetrazolium reduction assays has been demonstrated to be influenced by multiple factors [16] and the reaction was suggested to be taking place in the plasma membrane due to activity of trans-plasma membrane electron transport [17] instead of cytoplasm or mitochondria due to activity dehydrogenase enzymes. Moreover, inhibition of dehydrogenase activity may not always indicate a loss of viability as the treatment may alter enzymatic activity without affecting the number of living cells and it is suggested to use proper controls to compensate for the effects on metabolism [15]. There are also concerns regarding the Ki67 expression as an indicator of proliferation rate as the protein levels were demonstrated to be influenced by the time cells spent in G0 [18].

In this study we focused on how common cell culture methods were interpreted and whether they could provide sufficient evidence for related claims. However, we did not investigate how each experiment was designed or conducted. Since the strength of evidence depends on the robustness of the design, even if the method chosen is appropriate, it can provide reliable evidence only if it is carried out properly. General recommendations on good cell culture practices such as cell line identification, contaminant screening, proper use of controls (negative, positive and vehicle), appropriate handling and storage of assay components as well as information about the factors interfering assay results (such as cell confluency, media components and plastic materials) are covered by OECD guide GIVIMP [11]. Other recommendations we would like to mention include: proper data normalization (such as normalization for basal cell death in proliferation rate measurement), investigators being blind for the analyses [19, 20], proper data visualization (such as using logarithmic axis plots to demonstrate cell growth) and accounting interfering factors (such as artifacts in high-throughput assay systems [21]).

In this work we offered a nomenclature recommendation by which the most common claims in cell culture studies may be distinctively expressed. As the findings of our study indicate, we believe a more meticulously written materials and methods section and careful selection and utilization of cell culture methods are critical to increase overall quality of preclinical research.

## Supporting information

**S1 Data.**
(CSV)

## Acknowledgments

We would like to thank Yasemin SEVAL ÇELİK for her contributions in revising this manuscript. We would also like to extend our thanks to the reviewers, comments of whom has improved the quality of this paper substantially.

## Author Contributions

**Conceptualization:** Ali Burak Özkaya, Caner Geyik.

**Data curation:** Ali Burak Özkaya, Caner Geyik.

**Formal analysis:** Ali Burak Özkaya, Caner Geyik.

**Investigation:** Ali Burak Özkaya, Caner Geyik.

**Methodology:** Ali Burak Özkaya, Caner Geyik.

**Project administration:** Ali Burak Özkaya.

**Resources:** Ali Burak Özkaya, Caner Geyik.

**Software:** Ali Burak Özkaya, Caner Geyik.

**Supervision:** Ali Burak Özkaya.

**Validation:** Ali Burak Özkaya, Caner Geyik.

**Visualization:** Caner Geyik.

**Writing – original draft:** Ali Burak Özkaya, Caner Geyik.

**Writing – review & editing:** Ali Burak Özkaya, Caner Geyik.

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
