## [Decision Letter · Decision Letter 0]

18 Jun 2021

PONE-D-21-12647

Evidence provided by high-impact cell culture studies does not support authors' claims

PLOS ONE

Dear Dr. Ôzkaya,

Thank you for submitting your manuscript to PLOS ONE. After careful consideration, we feel that it has merit but does not fully meet PLOS ONE’s publication criteria as it currently stands. Therefore, we invite you to submit a revised version of the manuscript that addresses the points raised during the review process.

It is important to assure that you reply to all concerns by the reviewers (especially Reviewers 3 and 4). Please note that the review provided by Reviewer 1 pertains only to ‘statistical aspects’ of the study and so does not comment on the ‘clinical aspects’ [like medical importance, relevance of the study, ‘clinical significance and implication(s)’ of the whole study, etc.] of the manuscript.

We look forward to receiving your revised manuscript.

Kind regards,

Hamidreza Montazeri Aliabadi

Academic Editor

PLOS ONE

2. Please consider posting the list of studies and their analysis as supporting information. If materials, methods, and protocols are well established, authors may cite articles where those protocols are described in detail, but the submission should include sufficient information to be understood independent of these references (https://journals.plos.org/plosone/s/submission-guidelines#loc-materials-and-methods).

Reviewers' comments:

Reviewer's Responses to Questions

**Comments to the Author**

1. Is the manuscript technically sound, and do the data support the conclusions?

Reviewer #1: Yes

Reviewer #2: Yes

Reviewer #3: No

Reviewer #4: Partly

2. Has the statistical analysis been performed appropriately and rigorously? 

Reviewer #1: Yes

Reviewer #2: Yes

Reviewer #3: No

Reviewer #4: Yes

3. Have the authors made all data underlying the findings in their manuscript fully available?

Reviewer #1: Yes

Reviewer #2: Yes

Reviewer #3: No

Reviewer #4: Yes

4. Is the manuscript presented in an intelligible fashion and written in standard English?

Reviewer #1: Yes

Reviewer #2: Yes

Reviewer #3: Yes

Reviewer #4: No

5. Review Comments to the Author

Reviewer #1: Important note: This review pertains only to ‘statistical aspects’ of the study and so ‘clinical aspects’ [like medical importance, relevance of the study, ‘clinical significance and implication(s)’ of the whole study, etc.] are to be evaluated [should be assessed] separately/independently. Further please note that any ‘statistical review’ is generally done under the assumption that (such) study specific methodological [as well as execution] issues are perfectly taken care of by the investigator(s). This review is not an exception to that and so does not cover clinical aspects {however, seldom comments are made only if those issues are intimately / scientifically related & intermingle with ‘statistical aspects’ of the study}. Agreed that ‘statistical methods’ are used as just tools here, however, they are vital part of methodology [and so should be given due importance].

COMMENTS: In my opinion, this authors group [now there are only two but may include few more similar minded scientists working in same field] can very-well start developing ‘publishing guidelines’ for such articles/literature {I guess, that will be very useful (cell culture studies will likely support authors’ claims subsequently) and definitely will be a great contribution}. It seems that OECD’s ‘Guidance Document on Good In Vitro Method Practices (GIVIMP)’ is not sufficient. Is not that true? There are only nine articles quoted [included in reference, may be because only these many are available] but are good (very relevant) articles. Congratulations.

For this purpose {developing ‘publishing guidelines’ for such articles/literature }, excellent article quoted as reference-1, [Ioannidis JPA. Why Most Published Research Findings Are False. PLOS Med. 2005;2: e124. 260 doi:10.1371/journal.pmed.0020124] may be useful. The understudied aspect of this quality issue in published work highlighted is “the harmony between the hypotheses and the experimental design” is true for many areas and is appreciable. My only question is (line 52) ‘controls (positive, negative)’ {i.e. what are ‘positive controls and negative controls?}. Please clarify. Little/brief clarification of terms ‘p-hacking and cherry picking’ [like ‘P-hacking’ is the relentless analysis of data with an intent to obtain a statistically significant result, usually to support the researcher’s hypothesis and ‘Cherry-picking’ is the presentation of favourable evidence with the concealment of unfavourable evidence] could be useful for readers, I guess.

Although it is not my ‘Subject Area’, in my opinion, this one it is an excellent article. Nevertheless, I feel that overall the article is little short. Maybe adding few more case studies will do. As pointed out in ‘important note’ above “This review pertains only to ‘statistical aspects’ of the study and so ‘clinical aspects’ should be assessed separately/independently. May think of changing title (may make it more catchy).

Reviewer #2: In general, this is a well-conducted and important paper. The Authors are to be congratulated for their attempt to provide a degree of rigor that is typically lacking in preclinical cell culture studies.

In this study, the Authors examined highly-cited cell culture studies and investigated whether the claims made in each study were supported by sufficient experimental evidence.

To do this, they examined 282 claims asserted by 103 different high-impact articles. They found that many of all claims were not sufficiently supported by the evidence that was presented. They also identified troubling examples of misinterpretation of data.

They concluded there was “an alarming discordance” between the actual experimental findings and the way they were actually described in the manuscript.

To perform their analysis, the investigators searched on the key words “cytotoxicity, viability, cell death, growth inhibition, proliferation, or anti-cancer” and retrieved studies from 2017, 2018. They selected 121 publications, each of which had received at least 65 citations. Reviews and studies that did not include a claim or did not involve cell culture were excluded. This left 103 original studies for assessment. In total, the articles that were analysed were cited more than 9,000 times as of March 2021 (within two to four years of publication), so it is reasonable to assume (as the Authors have done) that despite their deficiencies, these publications have had an impact on the field.

The most useful aspects of this work include:

a) Authors provide a useful definition for each of the terms on which they searched - “cytotoxicity, viability, cell death, growth inhibition, proliferation, or anti-cancer”.

b) In Table 1 they provide a clear statement as to what constitutes sufficient and insufficient evidence for each of the terms that they defined. This is a valuable contribution as many publications do not apply the appropriate level of rigor when attempting to perform and interpret these assays.

c) The Authors also present specific proposals to improve the quality of future studies.

It would be a valuable contribution to preclinical research if the Authors’ proposals were widely adopted. This would certainly contribute to an improvement in the robustness of preclinical cell culture experiments.

Areas for improvement.

Major issues.

1) It would be helpful for the Authors to clarify their intent. Specifically, it is not clear to this Reader whether the Authors are stating that 64% of studies were supported with sufficient evidence, or 64% were not supported by sufficient evidence.

The two lines that appear to be in conflict are:

Line 33: “Our findings revealed that only 64% of all claims were sufficiently supported by evidence”

versus

Line 195: “we considered the evidence …which was asserted by 66 different studies (64%), as insufficient”

2) Two areas that were not specifically addressed, but that could further strengthen the manuscript are:

a) The Authors did not mention that subjective analyses should be performed by blinded investigators. Because it is seldom the case that these type of experiments are performed by blinded researchers, it would be valuable for the Authors to highlight its importance.

b) When considering an effect on cell proliferation or cell death, the Authors should stipulate that cell proliferation or cell death should be expressed on a logarithmic not a linear axis. Clearly cell proliferation is an exponential, not a linear function. Although many papers misrepresent their data by presenting it on a linear axis, where it looks more dramatic, this is not an appropriate way to present the data. Given the focus of this publication, this too would be a valuable addition for the Authors to comment upon.

Minor issues:

Line 162: Blc-2 should be Bcl-2

Reviewer #3: PONE-D-21-12647

The authors take on an important reproducibility issue in high-impact cell culture studies. This is certainly of high importance, and suggestions are made for improvements. Overall, the authors comment on a very important point, and one that is close to my own heart… viability vs cytotoxicity. However, they do not do this in a rigorous manner, and this is more suited for a commentary than any sort of research article or review.

1. Line 54 – is this substantiated by any references, or anecdotal?

2. The introduction and overall manuscript is lacking references

3. Lines 67+ reads more like commentary – also a bit vague. What is being discussed here? The cytotoxicity and cell viability terms?

4. What was the rationale for focusing on 2017 and 2018?

5. The inclusion/exclusion should be detailed more clearly on lines 91+

6. Since care is being taken to describe these cell health variables… viability I find a bit problematic. It is not the number of living cells, it generally represents a reaction that is often considered to have a linear relationship to the number of living cells. These enzymes, such as tetrazolium salt, etc., can be modified by exogenous factors and cannot be explicitly considered to describe living cell number.

7. Line 144 – LDH release is a cell viability assay, not a cytotoxicity assay – given the context of this paper, this is a very important point to classify these assays correctly. There are a variety of contexts described in the literature where cells leak LDH and remain otherwise alive and will respond normally via other cell viability measures. Adipocytes is one example of this, though others have been described. Unless the specific experimental context is being taken into account, it’s hard to see how these sorts of generalities are not going to adversely influence results here.

8. Table 1 – how is DNA content not included in cytotoxicity? It is the gold standard for determining cell number. You could say that it is context dependent, if there are no wash steps in a particular assay, but it cannot just be ignored as a cytotoxicity measure.

9. Given the well-described artifacts and issues with a lot of the cell viability and proliferation assays, as described by Hsieh et al., 2015, 2016, etc., this article seems to ignore many of these issues with the assays they included.

10. There doesn’t seem to be any transparency about how and why individual publications were deemed sufficient or insufficient. If highly cited publications are going to be published as insufficient, a rationale column for the decision-making process is critical to providing clarity for readers.

11. Lines 208+ - cytotoxicity is not a measure of viability, it is a different measure completely.

12. Line 206 – “we had to make assumptions on behalf of the researchers” is not very scientific. In a systematic review context, you can and should reach out to the authors for clarification. Appreciate this is not that, but if you are publishing a paper blasting the authors for having an “insufficient” study, and only rely on your interpretation, that is a problem.

13. Moreover, interpretation of who? Is this a single author performing these assumptions? Is each paper reviewed by both authors independently? A single person evaluating these is not robust – and what sort of testing was done to verify that the authors performed these assumptions in a consistent manner?

14. I feel at this point in the paper that I’m not sure of the overall goal. Are the authors just using the assay reported in the paper and checking which box on a table it fits into? This doesn’t seem like the level of discourse that is particularly worthy of a peer-reviewed publication. A commentary could fit this, but it seems more opinion than a rigorous evaluation of reproducibility – which is, of course, crucially important.

15. The first sentence of the discussion is very vague and alarmist – the only thing examined here was the cell proliferation/toxicity/viability and not any other claims these publications may have made. That is very important to make clear

Reviewer #4: The manuscript “Evidence provided by high-impact cell culture studies does not support authors’ claims” from Ozkaya and Geyik describes the use of various assays to evaluate cell state in published articles. The authors mined the literature for highly cited papers from 2017-2018 that evaluated cell viability, death, proliferation, growth, apoptosis, and cytotoxicity in cell culture systems, and considered whether or not the assay was appropriate and sufficient for the conclusions drawn. A summary of the authors’ ‘Evidence Sufficiency’ meta-analysis is presented in a single figure. They suggest that the incorrect use of assays, or over interpretation of results from specific assay classes are likely factors in the irreproducibility of preclinical studies.

Overall, the goal of the manuscript fits under the ‘Systematic Review and Meta-Analyses’ article category; however, it will require significant revisions and additional review should it be considered for publication. I cannot recommend it be published in its current form.

Please address the following points:

Minor

1. Add a citation to papers that have addressed similar issues – line 56-57.

2. Fig. 1C dehydrogenase activity is always used correctly to assess viability?

3. Fig. 1 C (discussed on lines 219-220) you say data are interpreted correctly in 73 claims, but the figure looks like it should be 48 claims. Please correct.

4. Thorough copy-editing is needed

Major

5. Nomenclature recommendations should not be buried in the Methods. You highlight this as a main conclusion of your work. There is an opportunity to be prescriptive with this work and for it to be a resource for people to turn to when selecting assays and designing experiments – I think this should be focused on much more.

6. Your definition of proliferation rate is presented as a population-level readout “how fast a group of cells divide over time” however it is then contradicted by stating that this rate only applies to surviving cells – an apparent decrease in proliferation rate of a population could occur from a slowing of the doubling time of all cells in a population or from a fractional response in which a subpopulation is killed off, and the surviving cells continue to proliferate at the same rate (or at an altered rate) resulting in a longer apparent doubling time, as you point out. Is your proposed definition of proliferation rate meant to be applied to single cell or population level data? Does it only apply to living cells at endpoint? If the endpoint is simply how fast a group of cells divide over time, then it doesn’t matter if the underlying phenotypes are different, though they should be determined with secondary assays designed specifically to measure cell death and growth rate.

7. I strongly disagree that measuring metabolic activity is sufficient for assaying cell viability. In some cases, these techniques (MTT, CellTiter-Glo, etc.) yield results that correlate linearly with live cell count, but many perturbations directly affect the metabolic activity of cells which can result in critical miss-interpretations of effects. See https://pubmed.ncbi.nlm.nih.gov/31302153/ for clear examples.

8. With respect to defining cell death, I generally agree that this needs to be measured directly, however, if the number of cells are counted at the time of drug addition, for example, then a treatment resulting in net cell loss can only result from some degree of cell death. In general, any time cells are counted, live and dead cells should be distinguished at the bare minimum. See Hafner et al. (https://pubmed.ncbi.nlm.nih.gov/27135972/) for a more nuanced and thorough discussion. Growth rate inhibition metrics, an important addition to the field, are omitted and should be discussed – they are immediately relevant and address several of the same concerns the authors highlight.

9. Apoptosis, please correct typo in “molecular switch responsible in apoptosis” to “molecular switch resulting in apoptosis” and specify whether you’re referring to cleaved caspases and/or MOMP.

10. Please clarify why you don’t consider PI sufficient to assay membrane integrity but you do consider other dyes that are also membrane impermeable to be sufficient (lines 174-177) viability assays.

11. Table 1 as is, is confusing especially for the proliferation and viability entries. It would be much more useful if it was presented by claim and then assay, followed by what each assay actually measures, and what is required for that assay to be sufficient for that claim.

12. Only nine references are provided – there is a lot of literature that has been overlooked on assay choices and pitfalls that should be considered, and incorporated into the introduction and discussion sections.

6. PLOS authors have the option to publish the peer review history of their article (what does this mean?). If published, this will include your full peer review and any attached files.

Reviewer #1: No

Reviewer #2: No

Reviewer #3: No

Reviewer #4: No

---

## [Author Response · Author response to Decision Letter 0]

23 Jul 2021

Authors: We would like to thank all reviewers for their constructive criticism. 

Reviewer #1

Authors: We would like to thank the reviewer for encouraging comments. There is indeed a lack of a comprehensive guideline for basic cell culture research especially regarding simple assays and how to interpret their results. I hope we can eventually start developing such guideline. We will seek likeminded researchers for this purpose as the reviewer has suggested. 

Specific points:

R1: My only question is (line 52) ‘controls (positive, negative)’ {i.e. what are ‘positive controls and negative controls?}. Please clarify. Little/brief clarification of terms ‘p-hacking and cherry picking’ [like ‘P-hacking’ is the relentless analysis of data with an intent to obtain a statistically significant result, usually to support the researcher’s hypothesis and ‘Cherry-picking’ is the presentation of favourable evidence with the concealment of unfavourable evidence] could be useful for readers, I guess.

Authors: An explanation has been added for positive and negative controls (lines 53, 54) as well as p-hacking and cherry picking (lines 56, 57). 

R1: May think of changing title (may make it more catchy).

We did try to find a catchier title when we first wrote the manuscript, but we did not like the titles we came up with.

Reviewer #2

Authors: Thank you for the reviewer’s comments and support. We tried to be as transparent and as clear as possible when writing the manuscript. Therefore, we really appreciate that our efforts did not go unnoticed. 

Specific points

R2: 1) It would be helpful for the Authors to clarify their intent. Specifically, it is not clear to this Reader whether the Authors are stating that 64% of studies were supported with sufficient evidence, or 64% were not supported by sufficient evidence. The two lines that appear to be in conflict are: Line 33: “Our findings revealed that only 64% of all claims were sufficiently supported by evidence” versus Line 195: “we considered the evidence …which was asserted by 66 different studies (64%), as insufficient”

Authors: Those statements are not in conflict, but we can understand the confusion. The first one was the ratio of sufficient claims (180/282). Second one was the ratio of studies with insufficient evidence (66/103). The number being same is just a coincidence. However, during revision we made minor changes in the analysis considering the feedback from the reviewers altering claim sufficiency rate to 63% (178/281) (line: 33). The number of studies with insufficient evidence stayed the same (line: 207) 

2) Two areas that were not specifically addressed, but that could further strengthen the manuscript are: a) The Authors did not mention that subjective analyses should be performed by blinded investigators. Because it is seldom the case that these type of experiments are performed by blinded researchers, it would be valuable for the Authors to highlight its importance. 

b) When considering an effect on cell proliferation or cell death, the Authors should stipulate that cell proliferation or cell death should be expressed on a logarithmic not a linear axis. Clearly cell proliferation is an exponential, not a linear function. Although many papers misrepresent their data by presenting it on a linear axis, where it looks more dramatic, this is not an appropriate way to present the data. Given the focus of this publication, this too would be a valuable addition for the Authors to comment upon.

Authors: We added a section containing relevant information to the discussion (lines: 280+). We included the importance of blind investigators as well as the importance of logarithmic scale for proliferation studies. 

R2: Minor issues: Line 162: Blc-2 should be Bcl-2 

Authors: Corrected.

Reviewer #3

Authors: Thank you for your valuable suggestions. We hope that the article is now more rigorous after this revision. 

Specific points:

R3: 1. Line 54 – is this substantiated by any references, or anecdotal? 

Authors: The references are listed in the previous sentence. Especially by the work of Ioannidis and his colleagues. We originally did not want to repeat the same references in two consecutive sentences. However, in accordance with the comments, we added them to the second sentence as well. 

R3: 2. The introduction and overall manuscript is lacking references

Authors. Manuscript was revised and new references were added.

R3: 3. Lines 67+ reads more like commentary – also a bit vague. What is being discussed here? The cytotoxicity and cell viability terms? 

Authors: The idea of this study emerged thanks to one simple observation: there were studies using tetrazolium reduction assay results to justify different claims such as viability, cytotoxicity etc. The core of this work is about whether these “claims” are properly sported by “evidence”. Therefore, we tried to clearly define terms corresponding to different claims. For this purpose, we first searched the literature, utilized the OECD guide and finally wrote our own definition. We believe it is important for our readers to understand our process clearly and fully. That is why that section was added. 

R3: 4. What was the rationale for focusing on 2017 and 2018?

Authors: We began our analysis in March 2020. We wanted to include a two-year period, we wanted to give the articles one year to receive citations (therefore we did not investigate 2018-2019) and we wanted to focus on the most recent work. 

R3: 5. The inclusion/exclusion should be detailed more clearly on lines 91+

Authors: We re-wrote the part. 

R3: 6. Since care is being taken to describe these cell health variables… viability I find a bit problematic. It is not the number of living cells, it generally represents a reaction that is often considered to have a linear relationship to the number of living cells. These enzymes, such as tetrazolium salt, etc., can be modified by exogenous factors and cannot be explicitly considered to describe living cell number.

Authors: We agree with the reviewer on the issue. Metabolic activity is not a very good indicator of cell viability. This is especially problematic when your treatment also modifies cellular metabolism. Imagine that you have developed a drug targeting mitochondrial metabolism. It is clear that you should not use tetrazolium reduction assay to evaluate cell viability. However, there are many studies doing exactly that. That is why it is important to address these issues. The same thing can be said for membrane integrity being an indicator of cell death, caspase-3 activation an indicator of apoptosis etc. However, in this study we did not want to focus on that because: even though tetrazolium reduction assay may not be an accurate indicator of viability, it is most definitely not an indicator of apoptosis. Yet there are highly-cited articles claiming just that. They say that the treatment induced apoptosis by providing only MTT results as evidence. This was the clearer and the bigger problem and we focused on that. However, we do take the comments of the reviewers into account and therefore we added a section to the discussion focusing on the problems pointed out (lines: 280+).

R3: 7. Line 144 – LDH release is a cell viability assay, not a cytotoxicity assay – given the context of this paper, this is a very important point to classify these assays correctly. There are a variety of contexts described in the literature where cells leak LDH and remain otherwise alive and will respond normally via other cell viability measures. Adipocytes is one example of this, though others have been described. Unless the specific experimental context is being taken into account, it’s hard to see how these sorts of generalities are not going to adversely influence results here.

Authors: Literature may seem to disagree with the reviewer on this (1-3). LDH assay almost always has been used to measure cytotoxicity. However, we agree that we should question validity of using membrane integrity as a measure of cell death. 

References: 

1. An enzyme-release assay for natural cytotoxicity. C Korzeniewski, D M Callewaert. 1983

2. Validation of an LDH assay for assessing nanoparticle toxicity. Xianglu Han et al. 2011. 

3. Cytotoxicity Assays: In Vitro Methods to Measure Dead Cells. Terry Riss et al. 2019

R3: 8. Table 1 – how is DNA content not included in cytotoxicity? It is the gold standard for determining cell number. You could say that it is context dependent, if there are no wash steps in a particular assay, but it cannot just be ignored as a cytotoxicity measure.

Authors: Table 1 was re-constructed. We did not form the table from out understanding of these assays and the evidence they provide. We formed the table from the data we obtained from the studies we have investigated. Therefore, if there is no study utilizing DNA content as a measure of cytotoxicity we cannot have it there. It is not a general statement; it is just a re-organization of the data we obtained from our analyses. However, since table was criticized for being not clear enough, we created another table and include a more detailed explanation to how we have analyzed the studies and created the table. 

R3: 9. 9. Given the well-described artifacts and issues with a lot of the cell viability and proliferation assays, as described by Hsieh et al., 2015, 2016, etc., this article seems to ignore many of these issues with the assays they included.

Authors: We included one of the studies as the reviewer suggested.

R3: 10. There doesn’t seem to be any transparency about how and why individual publications were deemed sufficient or insufficient. If highly cited publications are going to be published as insufficient, a rationale column for the decision-making process is critical to providing clarity for readers. 

Authors: We re-constructed table 1 to draw a clearer picture of our evaluation process. We included a rationale column as the reviewer suggested. 

R3: 11. Lines 208+ - cytotoxicity is not a measure of viability, it is a different measure completely.

Authors: We completely agree on cytotoxicity not being a measure of viability. However, we encountered articles using cytotoxicity as if it represents viability during our investigation. We included that section to acknowledge it. We do not defend it. 

R3: 12. Line 206 – “we had to make assumptions on behalf of the researchers” is not very scientific. In a systematic review context, you can and should reach out to the authors for clarification. Appreciate this is not that, but if you are publishing a paper blasting the authors for having an “insufficient” study, and only rely on your interpretation, that is a problem.

R3: 13. Moreover, interpretation of who? Is this a single author performing these assumptions? Is each paper reviewed by both authors independently? A single person evaluating these is not robust – and what sort of testing was done to verify that the authors performed these assumptions in a consistent manner?

Authors: There is no assumption and interpretation in the analysis part. The assumptions /interpretations were general and were carried out them before the analysis. The assumptions are listed as definitions in the materials and methods section. When we define viability, cytotoxicity, proliferation etc. we “assume” they the authors of the articles we are analyzing have used it the same way. But since we cannot know it for sure we included the section previously discussed (lines 217+)

We agree that out evaluation process was not clear enough in the original manuscript. We re-constructed Table 1 and included more text to clarify our process. 

R3: 14. I feel at this point in the paper that I’m not sure of the overall goal. Are the authors just using the assay reported in the paper and checking which box on a table it fits into? This doesn’t seem like the level of discourse that is particularly worthy of a peer-reviewed publication. A commentary could fit this, but it seems more opinion than a rigorous evaluation of reproducibility – which is, of course, crucially important.

Authors: We included more text to clarify our process. We tried out best to make the analysis as objective as possible. This study is not an evaluation of reproducibility. We focused on the claims and the evidence of the articles we have investigated. This study is neither a commentary nor a review. We had a hypothesis “highly cited articles do not have sufficient evidence to support their cell culture-related claims” and we focused on selected claims developed a strategy to test the hypothesis. We gathered data, made our analysis and have reached a conclusion. 

R3: 15. The first sentence of the discussion is very vague and alarmist – the only thing examined here was the cell proliferation/toxicity/viability and not any other claims these publications may have made. That is very important to make clear

Authors: The discussion was revised. 

Reviewer #3

Authors: We would like to thank the reviewer for helpful comments that substantially improved the quality of this manuscript. We hope that it could be considered for publication at its current stage. 

R4: Add a citation to papers that have addressed similar issues – line 56-57.

Authors: Citations added.

R4: Fig. 1C dehydrogenase activity is always used correctly to assess viability?

Authors: That is correct. In this study we considered dehydrogenase activity measurement to provide sufficient evidence for viability independent from the experimental design. On the other hand, we are well aware of metabolic activity not being a very accurate indicator of cell viability. We included a section in discussion to discuss it (lines 280+).

R4: 3. Fig. 1 C (discussed on lines 219-220) you say data are interpreted correctly in 73 claims, but the figure looks like it should be 48 claims. Please correct. 

Authors: Corrected. 

R4: 4. Thorough copy-editing is needed

Authors: Manuscript was revised accordingly. 

R4: 5. Nomenclature recommendations should not be buried in the Methods. You highlight this as a main conclusion of your work. There is an opportunity to be prescriptive with this work and for it to be a resource for people to turn to when selecting assays and designing experiments – I think this should be focused on much more.

Authors: That was our original intention. However, we realized that we needed those definitions in the materials and methods section to clearly explain our process. Considering the reviewers’ comments, we also included the definitions to discussion section and referred to them in the abstract. 

R4: 6. Your definition of proliferation rate is presented as a population-level readout “how fast a group of cells divide over time” however it is then contradicted by stating that this rate only applies to surviving cells – an apparent decrease in proliferation rate of a population could occur from a slowing of the doubling time of all cells in a population or from a fractional response in which a subpopulation is killed off, and the surviving cells continue to proliferate at the same rate (or at an altered rate) resulting in a longer apparent doubling time, as you point out. Is your proposed definition of proliferation rate meant to be applied to single cell or population level data? Does it only apply to living cells at endpoint? If the endpoint is simply how fast a group of cells divide over time, then it doesn’t matter if the underlying phenotypes are different, though they should be determined with secondary assays designed specifically to measure cell death and growth rate.

Authors: The main point is to differentiate slowing of the doubling time from killing a sub-population. We embraced that distinction to differentiate “proliferation rate” from “viability” and to give it a unique identity. We agree that the best way to measure it is to measure growth rate along with the cell death and we debated to decide what would be an acceptable evidence to demonstrate proliferation rate independent from cell death. Beside the usual markers which were specifically designed to measure proliferation rate, we decided “if there is no apparent reason for cell death (i.e. no treatment) and cells are being observed over time to observe how fast a group of cells divide over time, viability change can indicate proliferation”. However, it is important to note that while measuring proliferation-specific one must also consider the impact of cell loss due to death similar to the case with the viability. In essence, if cells die via cell death you may end up reading lower levels of Ki67 expression or nucleotide incorporation. That is why the proliferation data should also be normalized based on living cells. We included that recommendation to the discussion (lines: 280+). 

R4: 7. I strongly disagree that measuring metabolic activity is sufficient for assaying cell viability. In some cases, these techniques (MTT, CellTiter-Glo, etc.) yield results that correlate linearly with live cell count, but many perturbations directly affect the metabolic activity of cells which can result in critical miss-interpretations of effects. See https://pubmed.ncbi.nlm.nih.gov/31302153/ for clear examples.

We agree with the reviewer. We do not know how accurately metabolic activity indicates cell viability. This is especially problematic when your treatment also modifies cellular metabolism. Imagine that you have developed a drug targeting mitochondrial metabolism. It is clear that you should not use tetrazolium reduction assay to evaluate cell viability. However, there are many studies doing exactly that. That is why it is important to address these issues. The same thing can be said for membrane integrity being an indicator of cell death, caspase-3 activation an indicator of apoptosis etc. However, in this study we did not want to focus on that because: even though tetrazolium reduction assay may not be an accurate indicator of viability, it is most definitely not an indicator of apoptosis. Yet there are highly-cited articles claiming just that. They say that the treatment induced apoptosis by providing only MTT results as evidence. This was the clearer and the bigger problem and we focused on that. However, we do take the comments of the reviewers into account and therefore we added a section to the discussion focusing on the problems pointed out (lines: 280+).

R4: 8. With respect to defining cell death, I generally agree that this needs to be measured directly, however, if the number of cells are counted at the time of drug addition, for example, then a treatment resulting in net cell loss can only result from some degree of cell death. In general, any time cells are counted, live and dead cells should be distinguished at the bare minimum. See Hafner et al. (https://pubmed.ncbi.nlm.nih.gov/27135972/) for a more nuanced and thorough discussion. Growth rate inhibition metrics, an important addition to the field, are omitted and should be discussed – they are immediately relevant and address several of the same concerns the authors highlight.

Authors: We included discussion of these metrics (lines 270+).

R4: 9. Apoptosis, please correct typo in “molecular switch responsible in apoptosis” to “molecular switch resulting in apoptosis” and specify whether you’re referring to cleaved caspases and/or MOMP.

Authors: Corrected

R4: 10. Please clarify why you don’t consider PI sufficient to assay membrane integrity but you do consider other dyes that are also membrane impermeable to be sufficient (lines 174-177) viability assays.

Authors: Thanks to the comments of the reviewer, we realized that we made a mistake evaluating the use of membrane integrity markers for measurement of viability. We now classify PI and AAD7 as insufficient. We only accepted trypan blue as sufficient because living cells can be detected via trypan blue assay. We re-calculated all the numbers and re-draw all the graphs accordingly. 

R4: 11. Table 1 as is, is confusing especially for the proliferation and viability entries. It would be much more useful if it was presented by claim and then assay, followed by what each assay actually measures, and what is required for that assay to be sufficient for that claim.

Authors: Table-1 is reconstructed to better reflect our evaluation process. 

R4: 12. Only nine references are provided – there is a lot of literature that has been overlooked on assay choices and pitfalls that should be considered, and incorporated into the introduction and discussion sections.

Authors: More references are added.

---

## [Decision Letter · Decision Letter 1]

19 Aug 2021

PONE-D-21-12647R1

Evidence provided by high-impact cell culture studies does not support authors' claims

PLOS ONE

Dear Dr. Özkaya,

Thank you for submitting your manuscript to PLOS ONE. After careful consideration, we feel that it has merit but does not fully meet PLOS ONE’s publication criteria as it currently stands. Therefore, we invite you to submit a revised version of the manuscript that addresses the points raised during the review process.

Please specially address the concerns raised by reviewers 3 and 4. It might be helpful, if you can revise the title of the manuscript, so it is more specific ad does not seem dismissive towards all "authors' claims". Also, the text could also be slightly revised to clarify that the objective is not to dismiss all the findings these publications are reporting.

We look forward to receiving your revised manuscript.

Kind regards,

Hamidreza Montazeri Aliabadi

Academic Editor

PLOS ONE

Journal Requirements:

Reviewers' comments:

Reviewer's Responses to Questions

**Comments to the Author**

1. If the authors have adequately addressed your comments raised in a previous round of review and you feel that this manuscript is now acceptable for publication, you may indicate that here to bypass the “Comments to the Author” section, enter your conflict of interest statement in the “Confidential to Editor” section, and submit your "Accept" recommendation.

Reviewer #1: All comments have been addressed

Reviewer #2: All comments have been addressed

Reviewer #3: (No Response)

Reviewer #4: (No Response)

2. Is the manuscript technically sound, and do the data support the conclusions?

Reviewer #1: Yes

Reviewer #2: Yes

Reviewer #3: Partly

Reviewer #4: Partly

3. Has the statistical analysis been performed appropriately and rigorously? 

Reviewer #1: Yes

Reviewer #2: Yes

Reviewer #3: N/A

Reviewer #4: (No Response)

4. Have the authors made all data underlying the findings in their manuscript fully available?

Reviewer #1: Yes

Reviewer #2: Yes

Reviewer #3: (No Response)

Reviewer #4: Yes

5. Is the manuscript presented in an intelligible fashion and written in standard English?

Reviewer #1: Yes

Reviewer #2: Yes

Reviewer #3: Yes

Reviewer #4: No

6. Review Comments to the Author

Reviewer #1: COMMENTS: As already said, that everything [like execution, methodology, statistical analyses, etc] are very good [i.e. study is excellent]. Writing/presentation is also excellent. Later, comments made [probably by other respected reviewers as well] were/are attended or answered positively/adequately, I am fully satisfied and the manuscript is improved a lot. I recommend acceptance.

Reviewer #2: Thank you for addressing the comments that I had provided.

This is a useful contribution and an important contribution.

Reviewer #3: I appreciate the reviewer edits to the manuscript, though many of my original concerns remain. This article still reads to me as a commentary - an important topic for one - but is not well enough defined for publication as a research article.

I still have major concerns with the title of this article. The authors rather blindly call out other papers yet (Evidence provided by high-impact cell culture studies does not support authors' claims) yet they have not examined anything in these papers other than the use of cytotoxicity/cell viability assays. This absolutely must be made crystal clear. I asked for this on first revision and this was not addressed. The inflammatory title, when this is explicitly not assessed, is not proper.

It also needs to be explicitly stated that the claims they are examining from these papers are in some cases only a very small part of the overall paper. The authors in the response to reviewer document repeatedly state the rationale for this article, but they have strayed far beyond that in the paper itself, as written.

I happen to disagree with the authors on the LDH assay, though appreciate there can be difference of opinion here. Though the irony of them citing the literature to defend correct interpretation is not lost on me, given the topic of their paper. I would suggest that they think about what this assay actually examines and whether that fits into the definition of a cell viability or toxicity assay. As there are clear examples where the cell type and context promote positive findings in this assay, it cannot be a cytotoxicity assay.

Some language editing is still needed here, particularly for the newly added sections highlighted in yellow.

This article should be a much shorter commentary, where it could accomplish the author's goals and not over-reach.

Again, how is DNA content listed under proliferation and not toxicity? This was not appropriately responded to in revision.

It is still unclear how some terms are being distinguished. On line 135, cytotoxicity, the definition specifically states "cell death". Yet in Table 1, cytotoxicity and cell death are separate groups on this table. I don't understand how these groups are being defined and how they are different, given the descriptions the authors are providing here.

Reviewer #4: PONE-D-21-12647_R1 review (also attached)

The revised manuscript “Evidence provided by high-impact cell culture studies does not support authors’ claims” from Ozkaya and Geyik is improved from the initial submission; however, several points still need to be addressed and clarified prior to publication. The expansion of Table 1 is a significant improvement to the previous version.

Minor points:

1. Table 1: I think it would be more intuitive to change the order slightly so the cell death claims (apoptosis, cell death, cytotoxicity) are followed by the live cell claims together (cell growth, proliferation, viability) rather than strictly listing them alphabetically.

2. Thorough copy editing is still required. Much of the text remains unchanged.

3. Referring to cell state assays as ‘cell culture techniques’ (for example on lines 41, 266 and 282) is confusing. I perceive good cell culture technique to be the proper maintenance of cells in culture (free of mycoplasma and other contaminants, log-phase growth for the assays you discuss etc). Good cell culture technique is also a critical factor in generating high quality and reproducible measures of cell viability/death/proliferation etc. It would be worth adding that all of the assays discussed can be badly confounded by poor cell culture technique. As you correctly point out, cost, simplicity, speed, and throughput likely motivate assay selection.

4. Reference to Fig 1D (line 226) should be to 1C.

5. While a valuable clinical biomarker, ki67 positivity as it relates to cell cycle and cells cycling is complicated in culture systems, and many cells in arrested states remain ki67 positive – caution in its interpretation is warranted, and like most of the endpoints discussed should be cross-validated. See https://www.ncbi.nlm.nih.gov/pmc/articles/PMC6108547/ and https://elifesciences.org/articles/13722

6. Lines 37-38 are unclear.

7. Consider reframing line 76 from ‘we decided to define these terms ourselves’ to ‘in the current work, we propose a series of definitions and recommendations’ since this is the ultimate goal.

8. Lines 132 and 133, please be specific about the ‘morphological changes’ and ‘molecular switches’ that you are referring to.

9. Please add the citations for id#9 and id#26 in the text (lines 260 and 261) so the reader can refer to them directly without having to access your web tool.

10. Line 271: GR refers to normalized growth rate inhibition, not simply growth rate.

11. Line 272: DIP should be specified ‘drug-induced proliferation rate’

12. Lines 273-276 are unclear. Please explain.

Major points:

1. Your claim in Table 1 that ‘DH activity correlates with the number of viable cells’ is not always true; at the very least, the associated ‘Yes’ should be changed to ‘Conditional’. Moreover, you twice assert that you consider measurements of metabolic activity as sufficient evidence for viability because they’re widely used: ‘…it is the most common method used to evaluate viability…’ (line 119) and ‘…we considered measurement of metabolic activity as an indicator of viability as it is by far the most preferred viability assay…’ (lines 283-284). This is poor justification for reasons you include in the discussion and reasons brought up in the previous reviews. With proper controls/assay calibration (how does the assay output relate to live cell count in the specific context?) and cross-validation experiments to confirm that metabolic activity is not influenced by the perturbation under study, these assays of metabolic activity can be used as surrogates for viability, but these controls are critical to correct data interpretation. My concern is that these points are not sufficiently addressed, and that incorrect use of metabolic assays to evaluate cell viability and proliferation is presented as acceptable without caveats throughout the body of the paper only to be contradicted in the discussion.

2. Can you comment on how often, in the papers you included, the assays you considered are the only line of evidence for a conclusion drawn since this seems to be the most problematic case? A common scenario is one where a crude, population level measure such as MTT is used to suggest toxicity, and then followed by a microscopy or flow cytometry-based assay to confirm cell death vs change in proliferation and mechanism (apoptosis vs necrosis etc). In a case like that, would the use of a metabolic assay be considered sufficient or insufficient? The use of complimentary and confirmatory assays should be discussed and considered when scoring claims made in the papers that you evaluated.

7. PLOS authors have the option to publish the peer review history of their article (what does this mean?). If published, this will include your full peer review and any attached files.

Reviewer #1: **Yes: **Dr. Sanjeev Sarmukaddam

Reviewer #2: No

Reviewer #3: No

Reviewer #4: No

---

## [Author Response · Author response to Decision Letter 1]

23 Sep 2021

Please specially address the concerns raised by reviewers 3 and 4. It might be helpful, if you can revise the title of the manuscript, so it is more specific ad does not seem dismissive towards all "authors' claims". Also, the text could also be slightly revised to clarify that the objective is not to dismiss all the findings these publications are reporting.

The title was changed, and we included several statements to different sections of the article to clarify the points the editor made.

Evidence provided by high-impact cell culture studies does not support authors’ claims

Reviewer #1: COMMENTS: As already said, that everything [like execution, methodology, statistical analyses, etc] are very good [i.e. study is excellent]. Writing/presentation is also excellent. Later, comments made [probably by other respected reviewers as well] were/are attended or answered positively/adequately, I am fully satisfied and the manuscript is improved a lot. I recommend acceptance.

Thank you.

Reviewer #2: Thank you for addressing the comments that I had provided.

This is a useful contribution and an important contribution.

Thank you.

Reviewer #3: I appreciate the reviewer edits to the manuscript, though many of my original concerns remain. This article still reads to me as a commentary - an important topic for one - but is not well enough defined for publication as a research article.

We believe that this work fits the definition of a “research article” embraced by PlosOne:

“Research Articles present the results of original research that address a clearly defined research question and contribute to the body of academic knowledge”

We have a clearly defined research question (are cell culture claims of high-impact publications sufficiently supported by experimental evidence), we have an experimental approach to test the hypothesis and obtained results. Finally, we believe that the findings of this study as well as our conclusions contribute to the body of academic knowledge. 

We leave the final decision on the matter to the editor. 

I still have major concerns with the title of this article. The authors rather blindly call out other papers yet (Evidence provided by high-impact cell culture studies does not support authors' claims) yet they have not examined anything in these papers other than the use of cytotoxicity/cell viability assays. This absolutely must be made crystal clear. I asked for this on first revision and this was not addressed. The inflammatory title, when this is explicitly not assessed, is not proper.

We have changed the title. 

It also needs to be explicitly stated that the claims they are examining from these papers are in some cases only a very small part of the overall paper. The authors in the response to reviewer document repeatedly state the rationale for this article, but they have strayed far beyond that in the paper itself, as written.

We have changed the title and included statements to the abstract, introduction, and discussion sections to clarify this. 

I happen to disagree with the authors on the LDH assay, though appreciate there can be difference of opinion here. Though the irony of them citing the literature to defend correct interpretation is not lost on me, given the topic of their paper. I would suggest that they think about what this assay actually examines and whether that fits into the definition of a cell viability or toxicity assay. As there are clear examples where the cell type and context promote positive findings in this assay, it cannot be a cytotoxicity assay.

After discussing the points made by the reviewer, we decided not to change our original position on the matter. Cytotoxic insult causes loss of membrane integrity and intracellular content such as LDH leak from death/dying cells. Measuring LDH activity in cell media correlates with the loss of membrane integrity thus, cytotoxicity. Like every indirect assay, it has its limitations, but those limitations are not enough to dismiss it. 

It is important to note that the literature we cited in our previous response were carefully selected as they explain why membrane integrity was an indicator of cell death providing experimental evidence. We want to make it clear that even if we criticize published work in our study, we still believe citing published work is one of the pillars of science. 

Some language editing is still needed here, particularly for the newly added sections highlighted in yellow.

Proofreading on manuscript was carried out by a native English speaker and changes were made accordingly.

This article should be a much shorter commentary, where it could accomplish the author's goals and not over-reach.

This has been answered in first point.

Again, how is DNA content listed under proliferation and not toxicity? This was not appropriately responded to in revision.

DNA content is listed under proliferation because that is how the articles we have investigated have used them. We did not decide which evidence supports which claims. That information was taken from the articles we have investigated. Since there was no article using DNA content to measure toxicity, we cannot have it in that list. 

In other words, we can summarize the methodology used as:

1- Find the studies that contains at least one claim (such as cytotoxicity, proliferation, etc.)

2- Look for the methods used to make “that” claim in those articles (such as DNA content, dehydrogenase activity, etc.)

3- Decide whether the method used by the article provides sufficient evidence for the claim asserted in the article. 

For example, when Study A claims cytotoxicity by using MTT, we included that to our analysis and decided whether it is sufficient or not.

Another example, when Study B claims proliferation by using DNA content, we included that to analysis and decided whether it is sufficient or not.

So, if any study among the investigated 103 studies have had claimed cytotoxicity by demonstrating DNA amount as evidence, we would have included that. And then we would have decided the sufficiency. However, there were no studies using DNA content as evidence for cytotoxicity. 

By the way, we also agree with the reviewer on DNA content measurement being an appropriate technique to detect cytotoxicity, it just was not used in any of the 103 studies we analyzed and that is why it was not on the table.

It is still unclear how some terms are being distinguished. On line 135, cytotoxicity, the definition specifically states "cell death". Yet in Table 1, cytotoxicity and cell death are separate groups on this table. I don't understand how these groups are being defined and how they are different, given the descriptions the authors are providing here.

The table was generated by using the information gathered from the articles we have investigated. So, if the authors of an article used the term “cell death” or “cytotoxicity” in their manuscript, we included them separately in the table. Therefore, different claims in the table do not correspond to different definitions, that is just the data obtained from the articles. In fact, we considered cytotoxicity and cell death the same in our analysis. Similarly, we also considered growth and proliferation the same as well. However, we have written them separately in the table because the table contains the claims asserted by the authors of the articles we have investigated, and we just copied whatever they have chosen to write in their manuscripts. On line 135, on the other hand, we have written the definitions we have embraced in this study. That is why it specifically states “cell death” under cytotoxicity. We consider them as being equal and we carried out our analysis accordingly. 

Reviewer #4: PONE-D-21-12647_R1 review (also attached)

The revised manuscript “Evidence provided by high-impact cell culture studies does not support authors’ claims” from Ozkaya and Geyik is improved from the initial submission; however, several points still need to be addressed and clarified prior to publication. The expansion of Table 1 is a significant improvement to the previous version.

Minor points:

1. Table 1: I think it would be more intuitive to change the order slightly so the cell death claims (apoptosis, cell death, cytotoxicity) are followed by the live cell claims together (cell growth, proliferation, viability) rather than strictly listing them alphabetically.

We have changed the order as recommended.

2. Thorough copy editing is still required. Much of the text remains unchanged.

Proofreading on manuscript was carried out by a native English speaker and changes were made accordingly.

3. Referring to cell state assays as ‘cell culture techniques’ (for example on lines 41, 266 and 282) is confusing. I perceive good cell culture technique to be the proper maintenance of cells in culture (free of mycoplasma and other contaminants, log-phase growth for the assays you discuss etc). Good cell culture technique is also a critical factor in generating high quality and reproducible measures of cell viability/death/proliferation etc. It would be worth adding that all of the assays discussed can be badly confounded by poor cell culture technique. As you correctly point out, cost, simplicity, speed, and throughput likely motivate assay selection.

Word “techniques” in mentioned lines was changed to “methods”.

4. Reference to Fig 1D (line 226) should be to 1C.

Thank you for the nice catch. We have corrected the figure reference.

5. While a valuable clinical biomarker, ki67 positivity as it relates to cell cycle and cells cycling is complicated in culture systems, and many cells in arrested states remain ki67 positive – caution in its interpretation is warranted, and like most of the endpoints discussed should be cross-validated. See https://www.ncbi.nlm.nih.gov/pmc/articles/PMC6108547/ and https://elifesciences.org/articles/13722

A part was added to the discussion to mention this. 

6. Lines 37-38 are unclear.

The part was re-written.

7. Consider reframing line 76 from ‘we decided to define these terms ourselves’ to ‘in the current work, we propose a series of definitions and recommendations’ since this is the ultimate goal.

We have reframed the sentence as recommended.

8. Lines 132 and 133, please be specific about the ‘morphological changes’ and ‘molecular switches’ that you are referring to.

The part was specified. 

9. Please add the citations for id#9 and id#26 in the text (lines 260 and 261) so the reader can refer to them directly without having to access your web tool.

Citations were added. 

10. Line 271: GR refers to normalized growth rate inhibition, not simply growth rate.

It was corrected. 

11. Line 272: DIP should be specified ‘drug-induced proliferation rate’

It was corrected. 

12. Lines 273-276 are unclear. Please explain.

The part was re-written.

Major points:

1. Your claim in Table 1 that ‘DH activity correlates with the number of viable cells’ is not always true; at the very least, the associated ‘Yes’ should be changed to ‘Conditional’. Moreover, you twice assert that you consider measurements of metabolic activity as sufficient evidence for viability because they’re widely used: ‘…it is the most common method used to evaluate viability…’ (line 119) and ‘…we considered measurement of metabolic activity as an indicator of viability as it is by far the most preferred viability assay…’ (lines 283-284). This is poor justification for reasons you include in the discussion and reasons brought up in the previous reviews. With proper controls/assay calibration (how does the assay output relate to live cell count in the specific context?) and cross-validation experiments to confirm that metabolic activity is not influenced by the perturbation under study, these assays of metabolic activity can be used as surrogates for viability, but these controls are critical to correct data interpretation. My concern is that these points are not sufficiently addressed, and that incorrect use of metabolic assays to evaluate cell viability and proliferation is presented as acceptable without caveats throughout the body of the paper only to be contradicted in the discussion.

We agree with the reviewer, and we have made minor revisions to emphasize these points. However, after many discussions we decided not to change our analysis. Every assay has its limitations, and we must draw the line somewhere. In this study we decided to consider metabolic activity as an acceptable indicator of cell viability. The reviewer is correct on “being widely used” to be a poor justification. However, it is not only about being a commonly used tool. Viability measurement is the only purpose of an MTT assay that is somewhat acceptable. Again, it may not be perfect to measure viability, but the problem is that people are using it to measure cell death which is arguably far worse than using it to measure viability. And that is what we wanted to focus on this study. 

We investigated randomly selected articles from our list to detect whether they have used controls to confirm effects on metabolism. None of the articles have done a proper assay calibration. Therefore, when we change the sufficiency of MTT from “yes” to “conditional”, it is instead going to be “no” for almost all the claims. We are simply not ready to consider MTT as insufficient evidence for viability. 

Another issue with that approach was about the way this article presents its findings. We wanted to keep it as simple as possible to reach a wider audience. When we dive into the detail level in which we consider MTT as conditional evidence for viability, almost all experimental results are going to be “conditional evidence”. Indeed, proper controls and normalization are important for every assay, but that approach would turn this work into an article about cell culture assays and their weaknesses as we won’t be able to classify them as sufficient/insufficient for specific claims. We know that our current approach is also reductionist, but the idea is being not as reductionist as the work we criticize. By doing that we are hoping to reach more scientists and if we can make a couple of researchers question their decision to claim cell death with MTT data, we consider it as a win. After that initial step we can move on to a more complicated approach. We hope that someday we can be a part of team which creates a comprehensive guideline (on viability/cytotoxicity/proliferation rate etc) by which the current methodology is dissected, evaluated and practical recommendations are made.

2. Can you comment on how often, in the papers you included, the assays you considered are the only line of evidence for a conclusion drawn since this seems to be the most problematic case? A common scenario is one where a crude, population level measure such as MTT is used to suggest toxicity, and then followed by a microscopy or flow cytometry-based assay to confirm cell death vs change in proliferation and mechanism (apoptosis vs necrosis etc). In a case like that, would the use of a metabolic assay be considered sufficient or insufficient? The use of complimentary and confirmatory assays should be discussed and considered when scoring claims made in the papers that you evaluated.

When we were evaluating the articles’ claims we focused on the strongest evidence. So, if the article claimed apoptosis and has MTT and Annexin-V as evidence, we did not list MTT for their apoptosis claim, we just listed Annexin-V and considered it sufficient. That approached kept our list simple and concise. We detailed our methodology it a bit more in materials and methods section to clarify this. 

However, if the claim specifically states MTT without mentioning further experimentation we focused on that. For example: article no4 claimed proliferation rate change and performed an MTT and nucleotide incorporation assay. However, since there is this statement “MTT assay was performed to assess cell proliferation” we included it in our list as a separate entry and considered it insufficient. 

Article no121 claimed viability, cytotoxicity, cell death and apoptosis. They performed MTT and Annexin-V assays. However, since they state “The cytotoxicity of mPEG–PCL micelle to MCF-7 cells was evaluated using MTT assays.” we included MTT as a separate entry in the list and considered it as insufficient. 

Like this examples in all of the cases we did not focus on what really happened to the cells in the experiments, instead we focused on how it was presented/discussed in the article. 

On the other hand, many articles were easier to analyze including article no9 claiming viability, cytotoxicity, apoptosis and proliferation rate inhibition. The only data they had: MTT. 

We rechecked articles with multiple evidence to detect any cases in which authors provide a better follow up evidence that we might have missed. During that investigation we decided that the article no26 has stronger evidence for anti-cell activity (anti leukemic cell activity, in their case). That evidence was in tissue level and that is why we have missed it in the first place. Consequently, it was removed from the claim list. We reanalyzed the data based on this alteration and changed relevant parts accordingly.

---

## [Decision Letter · Decision Letter 2]

28 Oct 2021

PONE-D-21-12647R2From viability to cell death: claims with insufficient evidence in high-impact cell culture studiesPLOS ONE

Dear Dr. Ozkaya,

I would like to start by apologizing for the lengthy review process. Your manuscript, as I am sure you know, is about a controversial subject that has triggered discussions among the reviewers. As you will see below, all reviewers agree that the latest version of the manuscript has improved significantly; however, while reviewers 1 and 2 both agreed that manuscript should be accepted, two reviewers (Reviewers 3 and 4), have cited a few minor points that requires your attention. Hopefully, they are not too time-consuming to answer, and this would be the final re-submission.

We look forward to receiving your revised manuscript.

Kind regards,

Hamidreza Montazeri Aliabadi

Academic Editor

PLOS ONE

Journal Requirements:

Reviewers' comments:

Reviewer's Responses to Questions

**Comments to the Author**

1. If the authors have adequately addressed your comments raised in a previous round of review and you feel that this manuscript is now acceptable for publication, you may indicate that here to bypass the “Comments to the Author” section, enter your conflict of interest statement in the “Confidential to Editor” section, and submit your "Accept" recommendation.

Reviewer #1: All comments have been addressed

Reviewer #2: All comments have been addressed

Reviewer #3: (No Response)

Reviewer #4: (No Response)

2. Is the manuscript technically sound, and do the data support the conclusions?

Reviewer #1: (No Response)

Reviewer #2: Yes

Reviewer #3: Partly

Reviewer #4: Partly

3. Has the statistical analysis been performed appropriately and rigorously? 

Reviewer #1: (No Response)

Reviewer #2: Yes

Reviewer #3: N/A

Reviewer #4: Yes

4. Have the authors made all data underlying the findings in their manuscript fully available?

Reviewer #1: (No Response)

Reviewer #2: Yes

Reviewer #3: Yes

Reviewer #4: Yes

5. Is the manuscript presented in an intelligible fashion and written in standard English?

Reviewer #1: (No Response)

Reviewer #2: Yes

Reviewer #3: Yes

Reviewer #4: No

6. Review Comments to the Author

Reviewer #1: COMMENTS: As already said, methodology, statistical analyses, etc are good and the manuscript is improved a lot. I recommend acceptance.

Reviewer #2: No additional comments - my previous comments have been adequately addressed.

Congratulations on a valuable study.

Reviewer #3: The authors have done a good job of responding to most reviewer concerns, though on re-review I have a few limited concerns. The bulk of the data from this study is provided in an alternate web link, though I'm not sure how I feel about this. Is this a stable link? Is the journal providing certainty that this will stay active in perpetuity? This paper becomes completely illegible without these supporting data. I strongly suggest they be added as supplemental data to the manuscript itself to ensure that these data are retained.

I still feel there are some lingering issues, including some raised by reviewer #4. The final question about multiple assays, I think, raises some concerns with this analysis. This addresses how articles using multiple assays or measures were treated. The author answer was that they judged the best measure and judged the article on that - though insufficient details are provided on how they judged the best assay for each specific paper and condition. Also, certainly some credit should be given to authors that couple multiple assays together - these can often provide a more complete picture of cell health than any one individual assay.

I also feel like there are some interesting concerns raised by reviewer 4 in regard to controls and the author answer to it. Surely the use of an assay, without appropriate controls, is not sufficient. Yet it seems that the details of the assay and the specifics of how it was performed and controlled for are not factored into these determinations. When making this still broad statement about the credibility of these author claims, this seems an important aspect to have considered.

Reviewer #4: The article revised by Ozkaya and Geyik is improved. In particular I appreciate the addition of discussion around the caveats of the study and assays discussed. However, I find that the article has gained length rather than clarity, and that the authors are still missing an opportunity to provide clear recommendations for good cell culture technique (as it pertains to all cell-based assays) and optimal assay use to address specific biological questions accurately. While I see value in drawing attention to widespread incorrect or over-interpretation from basic assays, the potential for impact lies in being explicit about what would have been needed for a claim to be deemed sufficient. This information is only included in the Methods and is incomplete (examples are provided), the authors could consider adding a summary table of assays and the conclusions that can be drawn from them. Although I will leave it to the editor to decide whether or not this falls within the scope of the current paper.

I continue to disagree with the authors’ assessment that ‘viability measurement is the only purpose of an MTT assay that is somewhat acceptable’, when really it is evaluating changes in metabolic activity in a population of cells. The same number of cells can have vastly different MTT readouts based on treatment, and while I agree that using these assays to assert cell death is particularly problematic, the oversimplification is detrimental to the paper. An increase in signal does not always indicate an increase in cell number.

Remove references to the discussion from the Methods section.

I do not understand why cell count and DNA content are only considered sufficient evidence for proliferation in the absence of treatment. Background levels of cell death need to be assayed under all conditions, controls included.

Why is ‘Real-time cell imaging’ (which I assume to refer to time-lapse microscopy) insufficient for quantifying cell death? Watching cells over time is both a straightforward and accurate way to monitor cell fate.

DAPI is membrane permeable, how is it correctly used to test membrane integrity? It can be used for DNA content, and typically is used to complement a membrane impermeable dye to accurately evaluate the fraction of cell death. Same concern for acridine orange and Hoechst in the Viability section of the table. Unless they are used with GhostDye (or PI etc), or DNA content is accounted for, they can only yield overall cell counts.

Please clarify the difference between ‘Dye inclusion’ and ‘Membrane integrity’.

Propidium iodine should be Propidium iodide.

pH3 positive cells represent a small fraction of a population and can represent cells arrested in M-phase, their presence alone does not reflect the overall proliferative potential of a population.

Change in proliferation is not limited to decrease in rate, perturbations can also stimulate proliferation.

The discussion needs to be re-worked and copy edited. In particular lines 297-310. The discussion of IC50 values does not seem to fit – although these methods aim to improve reproducibility of large-scale drug response studies, this is somewhat separate from the issues the current paper is trying to address.

7. PLOS authors have the option to publish the peer review history of their article (what does this mean?). If published, this will include your full peer review and any attached files.

Reviewer #1: **Yes: **Dr. Sanjeev Sarmukaddam

Reviewer #2: No

Reviewer #3: No

Reviewer #4: No

---

## [Author Response · Author response to Decision Letter 2]

10 Dec 2021

Reviewer #3: The authors have done a good job of responding to most reviewer concerns, though on re-review I have a few limited concerns. The bulk of the data from this study is provided in an alternate web link, though I'm not sure how I feel about this. Is this a stable link? Is the journal providing certainty that this will stay active in perpetuity? This paper becomes completely illegible without these supporting data. I strongly suggest they be added as supplemental data to the manuscript itself to ensure that these data are retained.

We have created an CSV document (excel file) and upload it as a supplementary data along with the manuscript.

Reviewer #3: I still feel there are some lingering issues, including some raised by reviewer #4. The final question about multiple assays, I think, raises some concerns with this analysis. This addresses how articles using multiple assays or measures were treated. The author answer was that they judged the best measure and judged the article on that - though insufficient details are provided on how they judged the best assay for each specific paper and condition. Also, certainly some credit should be given to authors that couple multiple assays together - these can often provide a more complete picture of cell health than any one individual assay.

When multiple methods are used, and different types of evidence are provided by a study we focused on the best one. Here when we say “best” what we mean is the best according to the table 1. For example, let’s say researchers have performed assays A, B and C to evaluate cytotoxicity. And let’s say, according to the table 1, A and B is “insufficient” while C is “sufficient”. In that case we just included “C” and ignored the use of “A” and “B” as they have more of a supportive role. Basically, if there were multiple types of evidence supporting a claim we only focused on “sufficient” ones, if there are any. A simple example would be apoptotic markers. Bcl and Bax expressions are usually measured along with DNA fragmentation or phosphotidyl-serine exposure. If we were to evaluate those separately, we should classify them as insufficient. But since there is a more relevant and sufficient evidence presented along with them, we exclude them from our analysis. We do agree that utilizing multiple assays would generate better evidence but we think that evaluating that would be out of scope of this work. 

Reviewer #3: I also feel like there are some interesting concerns raised by reviewer 4 in regard to controls and the author answer to it. Surely the use of an assay, without appropriate controls, is not sufficient. Yet it seems that the details of the assay and the specifics of how it was performed and controlled for are not factored into these determinations. When making this still broad statement about the credibility of these author claims, this seems an important aspect to have considered.

Although we agree that the proper use of controls is crucial to obtain good evidence, it is again not the focus of this study. This work is planned and conducted to demonstrate the misuse and misinterpretation of some cell culture methods and its high prevalence even in high impact articles. It is not however, designed as a guide on how to design cell culture studies, perform cell culture methods, or a document about good cell culture practices. With each review we fear that the manuscript is moving away from its intended original structure and aim. Our assessments were not about the credibility of the claims, we just want to point out whether a method, if done properly, can be utilized to measure something or not. Obviously, there are many factors affecting the overall reliability of your findings including: the number of samples, how the components of the assays as well as the controls were prepared, where and how did the samples and assay components were stored, was the cell line authenticated, was the right carrier control, right blind, right positive control, right negative control used, what was the room temperature, did antibiotics affect the findings etc. We are not trying to evaluate validity of the work we investigated, and we are not trying to assess their quality. That is why we focused on the claims and methods. However, there were a few exceptions where we had to investigate the experimental design itself. All those were listed as “conditional” in the table 1. If we were to look into the details experimental design and question the experimental conditions for each, then every entry would be “conditional”, and main points we are trying to make becomes lost. However when we re-write parts of the discussion section according to the recommendations of the reviewers we included a condensed general good practices recommendations (paragraph: 293-306). 

Reviewer #4: The article revised by Ozkaya and Geyik is improved. In particular I appreciate the addition of discussion around the caveats of the study and assays discussed. However, I find that the article has gained length rather than clarity, and that the authors are still missing an opportunity to provide clear recommendations for good cell culture technique (as it pertains to all cell-based assays) and optimal assay use to address specific biological questions accurately. While I see value in drawing attention to widespread incorrect or over-interpretation from basic assays, the potential for impact lies in being explicit about what would have been needed for a claim to be deemed sufficient. This information is only included in the Methods and is incomplete (examples are provided), the authors could consider adding a summary table of assays and the conclusions that can be drawn from them. Although I will leave it to the editor to decide whether or not this falls within the scope of the current paper.

We just want to point out again that the aim of this work is not to provide readers with recommendations for good cell culture technique. Although we agree that the article has gained length rather than clarity, it is mostly because the reviewers wanted to turn this work into something we never intended it to be. Consequently, with each revision, we are just trying to change the manuscript based on the recommendations without altering the core. And the core is just about drawing attention to widespread incorrect or over-interpretation of some basic assays. We are very grateful for the invaluable recommendations, and it is really a pleasure to see that there are researchers out there as enthusiastic as we are about in-vitro practices. However, we are neither ready nor willing to turn this work into a guide on good cell culture practices. 

We believe that the most important part of a scientific paper is the methods section. Placing the information regarding our approach on that section was a deliberate decision. We believe that methods part must contain all the information required to understand what was done in a study. Therefore, we placed both definitions we embraced along with our strategy to evaluate the claims in that section. The reviewer considers this information as incomplete because it does not include information regarding how to conduct and correctly interpret each method. However, we do not want to include those in this study. However, we did include some recommendations (paragraph: 293-306)

Reviewer #4: I continue to disagree with the authors’ assessment that ‘viability measurement is the only purpose of an MTT assay that is somewhat acceptable’, when really it is evaluating changes in metabolic activity in a population of cells. The same number of cells can have vastly different MTT readouts based on treatment, and while I agree that using these assays to assert cell death is particularly problematic, the oversimplification is detrimental to the paper. An increase in signal does not always indicate an increase in cell number.

We have made our points on this in the previous revision. We would like to leave the final decision on this to the editor. 

Reviewer #4: Remove references to the discussion from the Methods section.

References were removed.

Reviewer #4: I do not understand why cell count and DNA content are only considered sufficient evidence for proliferation in the absence of treatment. Background levels of cell death need to be assayed under all conditions, controls included.

We agree with the necessity of evaluating the basal level of death. However, studies we investigated did not measure background cell death when they claim proliferation. That is why we considered only the absence of treatment where the effects of cell death on the measurement is minimal, as acceptable. 

Reviewer #4 Why is ‘Real-time cell imaging’ (which I assume to refer to time-lapse microscopy) insufficient for quantifying cell death? Watching cells over time is both a straightforward and accurate way to monitor cell fate.

There was just one study where the “real-time cell imaging” was used to evaluate cell death (#49). In that study researchers just counted the number of viable cells over time. The graph was presented as “%viability” but they refer the data as an indicator of cell death. 

Reviewer #4 DAPI is membrane permeable, how is it correctly used to test membrane integrity? It can be used for DNA content, and typically is used to complement a membrane impermeable dye to accurately evaluate the fraction of cell death. 

DAPI can be membrane-permeable depending on the concentration and the incubation period but its rate of absorption by the cells is greatly increased in non-viable cells and fixated cells compared to viable ones. Therefore, it can be used to measure membrane integrity. 

1. https://www.ncbi.nlm.nih.gov/pmc/articles/PMC4496231/

2. https://pubmed.ncbi.nlm.nih.gov/27037070/

Reviewer #4 Same concern for acridine orange and Hoechst in the Viability section of the table. Unless they are used with GhostDye (or PI etc), or DNA content is accounted for, they can only yield overall cell counts.

In all the cases, acridine orange and Hoechst were used along with an indicator of cell death:

• Hoechst with PI (#60)

• Acridine orange with EtBr (#68)

• Acridine orange with PI (#77)

• Acridine orange with PI (#77)

• Acridine orange with DAPI (#77)

Reviewer #4 Please clarify the difference between ‘Dye inclusion’ and ‘Membrane integrity’.

Dye inclusion: methods utilizing dyes which are taken up by and indicate viable cells. Viable cell dyes.

Membrane integrity: methods utilizing dyes which can indicate the loss of membrane integrity. Non-viable cell dyes. 

Reviewer #4 Propidium iodine should be Propidium iodide.

Corrected

Reviewer #4 pH3 positive cells represent a small fraction of a population and can represent cells arrested in M-phase, their presence alone does not reflect the overall proliferative potential of a population.

In the study (#95) researchers accepted “lack of pH3 positive cells” as an indicator of halted proliferation. The logic is: “if there are no cells in M-phase in a group of cells, it means that the population is not proliferating.”. That is why we considered their evidence as sufficient. We included this explanation in Table 1. 

Reviewer #4 Change in proliferation is not limited to decrease in rate, perturbations can also stimulate proliferation.

We were not trying to claim that the change to be limited to the decrease. We have made changes in the manuscript to clarify this.

Reviewer #4 The discussion needs to be re-worked and copy edited. In particular lines 297-310. The discussion of IC50 values does not seem to fit – although these methods aim to improve reproducibility of large-scale drug response studies, this is somewhat separate from the issues the current paper is trying to address.

That paragraph almost entirely consists of sentences added as a response to the previous criticism provided by the reviewers. We agree that it is disconnected. We have re-written the part as suggested. IC50 part was removed.

---

## [Decision Letter · Decision Letter 3]

3 Jan 2022

From viability to cell death: claims with insufficient evidence in high-impact cell culture studies

PONE-D-21-12647R3

Dear Dr. Ozkaya,

We’re pleased to inform you that your manuscript has been judged scientifically suitable for publication and will be formally accepted for publication once it meets all outstanding technical requirements.

Kind regards,

Hamidreza Montazeri Aliabadi

Academic Editor

PLOS ONE

Additional Editor Comments (optional):

Reviewers' comments:

Reviewer's Responses to Questions

**Comments to the Author**

1. If the authors have adequately addressed your comments raised in a previous round of review and you feel that this manuscript is now acceptable for publication, you may indicate that here to bypass the “Comments to the Author” section, enter your conflict of interest statement in the “Confidential to Editor” section, and submit your "Accept" recommendation.

Reviewer #3: All comments have been addressed

2. Is the manuscript technically sound, and do the data support the conclusions?

Reviewer #3: Yes

3. Has the statistical analysis been performed appropriately and rigorously? 

Reviewer #3: N/A

4. Have the authors made all data underlying the findings in their manuscript fully available?

Reviewer #3: Yes

5. Is the manuscript presented in an intelligible fashion and written in standard English?

Reviewer #3: Yes

6. Review Comments to the Author

Reviewer #3: We appreciate the extensive revisions that have been completed by the author. While we continue to disagree on some of the finer points of this topic, these are discussions that can play out in the peer-reviewed literature. They have done a commendable job in ensuring all their data is available for any further discussions that other authors would like to have. I think this will be an interesting discussion point for further considerations.

7. PLOS authors have the option to publish the peer review history of their article (what does this mean?). If published, this will include your full peer review and any attached files.

Reviewer #3: No

---

## [Editor Report · Acceptance letter]

17 Jan 2022

PONE-D-21-12647R3 

From viability to cell death: claims with insufficient evidence in high-impact cell culture studies 

Dear Dr. Özkaya:

I'm pleased to inform you that your manuscript has been deemed suitable for publication in PLOS ONE. Congratulations! Your manuscript is now with our production department. 

Kind regards, 

on behalf of

Dr. Hamidreza Montazeri Aliabadi 

Academic Editor

PLOS ONE